# Microbial Phenolic Metabolites: Which Molecules Actually Have an Effect on Human Health?

**DOI:** 10.3390/nu11112725

**Published:** 2019-11-10

**Authors:** María Marhuenda-Muñoz, Emily P. Laveriano-Santos, Anna Tresserra-Rimbau, Rosa M. Lamuela-Raventós, Miriam Martínez-Huélamo, Anna Vallverdú-Queralt

**Affiliations:** 1Department of Nutrition, Food Science and Gastronomy, School of Pharmacy and Food Sciences and XaRTA, Institute of Nutrition and Food Safety (INSA-UB), University of Barcelona, 08921 Santa Coloma de Gramenet, Spain; mmarhuendam@ub.edu (M.M.-M.); elaversa21@alumnes.ub.edu (E.P.L.-S.); lamuela@ub.edu (R.M.L.-R.); avallverdu@ub.edu (A.V.-Q.); 2Consorcio CIBER, M.P. Fisiopatología de la Obesidad y Nutrición (CIBERObn), Instituto de Salud Carlos III (ISCIII), 28029 Madrid, Spain; anna.tresserra@iispv.cat; 3Departament de Bioquímica i Biotecnologia, Universitat Rovira i Virgili, Unitat de Nutrició Humana, Hospital Universitari San Joan de Reus, Institut d’Investigació Pere Virgili (IISPV), 43002 Reus, Spain

**Keywords:** microbiota, health, polyphenols, mass spectrometry, liquid chromatography, plasma, urine

## Abstract

The role of gut microbiota in human health has been investigated extensively in recent years. The association of dysbiosis, detrimental changes in the colonic population, with several health conditions has led to the development of pro-, pre- and symbiotic foods. If not absorbed in the small intestine or secreted in bile, polyphenols and other food components can reach the large intestine where they are susceptible to modification by the microbial population, resulting in molecules with potentially beneficial health effects. This review provides an overview of studies that have detected and/or quantified microbial phenolic metabolites using high-performance liquid chromatography as the separation technique, followed by detection through mass spectrometry. Both in vitro experimental studies and human clinical trials are covered. Although many of the microbial phenolic metabolites (MPM) reported in in vitro studies were identified in human samples, further research is needed to associate them with clinical health outcomes.

## 1. Introduction

The human gut is home to a diverse community of microorganisms, whose involvement in metabolism, disease, immunity, nutrition and an expanding number of other health-related issues is becoming increasingly evident. Their important role in the maintenance of the health-disease equilibrium is supported by studies reporting that germ-free animals are more likely to suffer infections and unbalanced nutrient uptake, and these animals only regain their health status when colonised by normal microbiota [1].

Each human has a unique gut microbiota that changes throughout life. The modifications begin at birth, when the microbiome is affected by the delivery method, and then by whether breast or formulated milk is given [2]. In subsequent years, the environment, diet and lifestyle all influence the microbiome [3,4]. Alterations in the microbiota can cause dysbiosis, which has been associated with a variety of health issues ranging from gastrointestinal diseases [5] to mental health [6] and aging [7,8]. The importance of maintaining a healthy microbiota has led to the development of a whole industry around pro-, pre- and even symbiotic foods [9,10].

It is not only the composition of the gut bacterial community, however, that is responsible for these changes in health. Food components that pass through the small intestine without absorption reach the colon, where they can be transformed by the microbiota into more easily absorbed metabolites with varying benefits [11]. The metabolic capacity of microorganisms in the gut has proven greater than that of the host [12,13] to the extent that the microbiota has been postulated to function as a separate organ [14].

Among the molecules that undergo microbial modification in the intestine are polyphenols, plant secondary metabolites that provide beneficial health effects through their interaction with the human body [15,16]. The main site of absorption of polyphenols is the small intestine, but when these molecules bear a glycoside group, they need to be hydrolysed prior to entering circulation [17]. This occurs through two possible mechanisms: the action of lactase-phlorizin hydrolase (LPH) or the cytosolic β-glucosidase (CBG). LPH, located at the edge of the enterocyte, deglycosylates the polyphenols, increasing their lipophilicity and allowing them to enter the enterocyte by passive diffusion. CBG is found inside the enterocyte and metabolises the more polar glycosides after they have been introduced through transporters such as sodium-dependent glucose transporter 1 SGLT1 [17].

Some polyphenols can reach the colon intact, or after being secreted in bile after following the enterohepatic cycle. In the gut, the microbiota can hydrolyse the glycosides and also transform the aglycones into new molecules [18]. Their transformation into simpler components involves different reactions, such as ring fission, hydrolysis, demethylation, reduction, decarboxylation, dehydroxylation and isomerisation [19,20]. There is evidence of bacterial presence in the small intestine; thus, microbial phenolic metabolites (MPM) could possibly be generated there [21,22]. However, investigations around microbial modification of food components have been mainly performed in colonic samples. This might be due to the high invasivity of small intestine sampling, but also because the food chyme spends less time there than in the large intestine, thus, microbes have less time to act on the polyphenols.

The phenolic compounds and metabolites absorbed in both the small intestine and the colon can cross the basolateral membrane of the enterocyte, where they can be locally conjugated into phase II metabolites (where they are transformed into glucuronides, sulphates and other conjugate forms), enter the bloodstream and reach the liver, where they undergo further conjugation [17,23]. Once they enter the systemic circulation, polyphenols can be distributed to most tissues, reaching even breast milk and the brain [24]. Finally, circulating polyphenols and their metabolites derived from human modification alone or after microbial modification are primarily excreted in urine [25,26]. Those polyphenols that have not been absorbed or reabsorbed are excreted through the faeces [23]. The cumulative excretion of MPM in urine provides a more accurate assessment of absorption than plasma pharmacokinetics. Treated as xenobiotics, MPM are rapidly removed from the bloodstream into the urine and their presence in the circulatory system is transitory [27].

All these processes are shown in Figure 1.

The analysis of MPM in biological samples is complex due to their structural diversity, their low and varied concentrations, the limited availability of standards, and the complexity of the biological matrix [24,28]. Plasma and urine are the most widely used human biological samples, although faeces, tissues, ileal fluid and breastmilk have also been studied [24].

MPM in plasma and urine samples have generally been studied using mass spectrometry (MS) methods, with differences in the separation mode [28]. Some studies have used gas chromatography (GC) or high-performance liquid chromatography (HPLC) and even ultra-high-performance liquid chromatography (UHPLC). Examples of these separation methods are found throughout this text. When the target compounds are conjugates, liquid chromatography (LC) is the preferred separation technique, because their low volatility renders GC unsuitable [28]. Additionally, due to the complexity of the matrix, some cleaning and extraction are sometimes performed to minimise noise [29].

The growing interest in microbiota and phenolic compounds has prompted researchers to explore their possible interactions. Moreover, phenolic compounds can act as prebiotics, modifying the growth of bacterial colonies, an effect mainly studied in vitro [30]. Conversely, the microbiota is able to modify phenolic compounds, leading to production of new molecules that can pass through the membrane of the enterocyte and impact human health [31].

These results could help researchers evaluate exposure to specific foods in clinical and epidemiological studies. However, because multiple molecules are tested at the same time, it is not possible to differentiate which metabolite comes from what molecule.

This review aims to summarise the findings of studies of polyphenol gut microbiota interactions using HPLC-MS to identify and/or quantify metabolites. The results of in vitro and/or in vivo studies are reported by phenolic group.

## 2. Human Interventions

### 2.1. Phenolic Acids

Coffee is one of the most popular beverages worldwide, and several epidemiological studies have linked its consumption to a decreased risk of type 2 diabetes, some cancers and cardiovascular disease. These health effects have been partly attributed to a high concentration of chlorogenic acids (CGAs). The low bioavailability of CGAs means their impact on health relies on metabolism by the gut microbiota, which has prompted investigations into this process.

Mills et al. [30] found that coffee with the highest levels of CGAs increased the growth of *Bifidobacterium spp.* 10 h after exposure. CGAs also increased *Clostridium coccoides–Eubacterium*, associated with the prevention of obesity and related diseases [32]. Ludwig et al. [33], after incubating human faecal samples with espresso coffee, identified and quantified 11 metabolites arising from CGA degradation by HPLC-MS and GC-MS. Caffeic and ferulic acids were the first to be formed after one hour of incubation. The major end products after six hours were dihydrocaffeic, dihydroferulic, and 3-(3’-hydroxyphenyl)propionic acids. The rate and extent of the breakdown were influenced by the faecal microbiota composition of the participants. These results were in accordance with a previous study, in which caffeic, chlorogenic and caftaric acids were incubated with human faecal microbiota and the metabolites identified by HPLC coupled to triple quadrupole MS. All free acids (caffeic, quinic and tartaric) were mainly metabolised to 3-hydroxyphenylpropionic and to a lesser extent to benzoic acid [34].

Gómez-Juaristi et al. [35] studied the absorption and metabolism of hydroxycinnamates after the consumption of 3.5 g of an instant soluble green and roasted coffee blend in 250 mL of hot water. Dihydrohydroxycinnamoylquinic acids (3-, 4- and 5-dihydrocaffeoylquinic acids, 3-, 4- and 5-dihydroferuloylquinic acids and dihydrocoumaroylquinic acid) were identified in urine for the first time after green coffee ingestion. Feruloylglycine and isoferuloylglycine were also detected, as well as phase II derivatives of lactones, namely three sulphated derivatives of caffeoylquinic lactone; one glucuronidated derivative of feruloylquinic lactone; 11 metabolic derivatives of hydroxyphenylpropionic, hydroxyphenylacetic, and hydrophenylbenzoic acids; and hydroxyhippuric acid and phloroglucinol.

The impact of colonic microbiota on the bioavailability of polyphenols from orange juice has also been studied [36]. After mixing and incubating target polyphenols with human faeces under anaerobic conditions, the phenolic acid metabolites were analysed by MS. Colonic microbiota had an impact on the production of hydroxy- and methoxyphenylpropionic acids, which are subsequently converted into hippuric acid and its hydroxylated counterparts in the liver. The bioactivity of such colon-derived metabolites is attracting growing interest. In vitro studies have indicated that protocatechuic, syringic, gallic and vanillic acids may play a protective role against atherosclerosis [37] and that 3-(3-methoxy-4-hydroxyphenyl)propionic acid has attenuating effects on diabetes [38].

### 2.2. Stilbenes

Resveratrol, the main stilbene found in grapes and red wine, is known for having anti-inflammatory effects, and an ability to block human platelet aggregation and promote eicosanoid synthesis [39]. Bode et al. [19] studied *trans*-resveratrol metabolism by the human gut microbiota in 12 healthy men. Twenty-four hours after of a single supplement of 0.5 mg trans-resveratrol/kg body weight, the microbial metabolites: dihydroresveratrol, lunularin and 3,4′-dihydroxy-*trans*-stilbene were found in urine.

### 2.3. Flavanones

The consumption of oranges or orange juice has been inversely correlated with ischemic stroke and acute coronary events, effects mainly attributed to the flavanone content of the fruit. Pereira-Caro et al. [26,27] identified 33 MPM in urine after the ingestion of orange juice, mainly cinnamic acids, phenylhydracrylic acids, phenylpropionic acids, phenylacetic acids, benzoic acids, mandelic acids, benzenetriols and hippuric acids. Maximum excretion was observed between 5–10 and 10–24 h after orange juice administration, indicating that all these compounds were derived from colonic microflora mediated flavanone metabolism. Ordóñez et al. [28] obtained similar results in a study where three healthy participants ingested 500 mL of orange juice, and urine was collected over a 24 h period. A total of 22 free phenolic acids, one benzenetriol and 35 phase II metabolites were identified and quantified. The MPM found in the study are listed in Appendix A.

### 2.4. Flavan-3-ols

Recent clinical trials have reported beneficial effects of cocoa polyphenols on cardiovascular health [40,41]. These include flavon-3-ols, present in significant amounts in cocoa mainly as monomers and polymers, and epicatechin and procyanidins [42]. However, the biological effects of flavon-3-ols depend on bioavailability. Epicatechin is easily absorbed in the small intestine and quickly metabolised, whereas polymeric procyanidins reach the colon, where they are transformed into hydroxyphenylvalerolactones by the gut microbiota. These metabolites are easily absorbed in the colonocyte and could have greater biological activity than the parent form [41,43].

In a feeding study by Martin et al. [44], 10 participants who regularly ate dark chocolate and 10 who rarely did so were given 2 × 25 g of dark chocolate daily and morning spot urine samples were collected each day. Metabolites of epicatechin (5-(3,4,dihydroxyphenyl)-γ-valerolactone, 2S-1-(3,4-dihydroxyphenyl)-3-(2,4,6-trihydroxyphenyl)-propan-2-ol and 4-hydroxy-5-(3´,4´-dihydroxyphenyl)-valeric acid)) were identified in both groups, but only participants who regularly ate chocolate saw an increase of sulphated, glucuronided and methyl-sulphated conjugates of 5-(3,4-dihydroxyphenyl)-valeric acid (Appendix A). These results suggest that gut microbial metabolism of cocoa phenolics differs according to the history of chocolate ingestion.

Urpí-Sardà et al. [42,45] also evaluated the consumption of cocoa. In this case, 42 healthy participants consumed 40 g/day of cocoa for four weeks. A total of 19 metabolites were identified and quantified in 24 h of urine collection, predominantly hydroxyphenylacetic acids and hydroxybenzoic acids. The principal groups of MPM in plasma were hydroxyphenylvalerolactones, hydroxyphenylpropionic acids, hydroxycinnamic acids and hydroxybenzoic acid (Appendix A).

Wiese et al. [43] described similar outcomes, where 5-(3′,4′-dihydroxyphenyl)-γ-valerolactone was the principal gut metabolite in plasma, and thus, the most relevant gut-mediated metabolite of procyanidin B1. The authors also confirmed interesting pathways of polymeric procyanidin degradation by gut microbiota. They suggested that some MPM arising from procyanidins, namely 5-(3´,4´-dihydroxyphenyl)-γ-valerolactone and 4-hydroxy-5(3’,4’-dihydroxyphenyl)valeric acid, may eventually be metabolised into conjugated 5-(3´,4´-dihydroxyphenyl)-γ-valerolactone and conjugated phenylvaleric acids (Appendix A).

Interestingly, Khymenets et al. [46] identified epicatechin metabolites derived from host and microbiota in human milk samples of nursing mothers after intake of dark chocolate. Human milk has a wide range of bioactive compounds interesting for human health [47]. The principal phase II host metabolites were methyl epicatequin, epicatechin glucuronide and sulphate. Dihydroxyphenylvalerolactone sulphate was the principal gut-mediated epicatechin metabolite detected in human milk.

Some studies have focused on flavanols from tea. After the consumption of 500 mL of green tea, several valerolactones were detected in urine in their conjugated form, linked to glucuronide, sulphate and methyl groups [25]. Duynhoven et al. [48] performed a study with black tea supplementation, and identified that host-conjugated metabolites derived from catechins, kaempferol and gallic acid appeared most rapidly in plasma (before one hour), compared to conjugated MPM such as valerolactones, valeric acids, phenols and phenolic acids (after two to four hours). However, due to the rapid appearance of MPM in plasma, the authors suggested that microbial activity could begin in the ileum and not in the colon. Likewise, while catechin conjugates continued in circulation for up to six hours, 5-(3ʹ,4ʹ-dihydroxyphenyl)-γ-valerolactone glucuronide and pyrogallol sulphate circulated at higher concentrations for up to almost 30 h. Schantz et al. [49] found relevant evidence that metabolism of green tea catechins and gallic acid may take place inside the ileum, similar to Duynhoven et al. [48]. Using an exvivo ileostomy model, (−)-epicatechin gallate was rapidly degraded to gallic acid and (−)-epicatechin during the first two hours of incubation. Gallic acid was further degraded into pyrogallol after 24 h of incubation. Likewise, 5-(3’,4’,5’-trihydroxyphenyl)-γ-valerolactone and 5-(3’,4’-dihydroxyphenyl)-γ-valerolactone were identified as metabolites of catechins and (−)-epicatechin, and the only metabolite of epicatechin gallate was 5-(3’,4’,5’-trihydroxyphenyl)-γ-valerolactone (Appendix A).

Inter-individual variation is another relevant factor to consider when assessing MPM bioavailability, as it may directly affect the biological response to polyphenol intake. Duynhoven et al. [48] reported that participants showed more variation in MPM such as valerolactones and valeric acids than in components directly absorbed in the small intestine such as catechins, after a single dose of black tea. Moreover, intake of polyphenol-containing foods can affect the gut microbiota profile and numbers, thereby increasing individual variability in metabolism [50]. Plasma levels of MPM can also be affected by long periods of supplementation. Clarke et al. [51] observed an increased concentration of valerolactones after three months of green tea supplementation.

Regarding the conjugate forms of MPM, it seems that the human metabolism has a stronger tendency towards sulphatation than glucuronidation. Van Duynhoven et al. [48] reported that plasma levels of sulphated pyrogallol and 5-(3ʹ,5ʹ-dihydroxyphenyl)-y-valerolactone were higher than the glucuronide forms. Similarly, Clarke et al. [51] found that the plasma valerolactone sulphate concentration was higher compared to other conjugated forms after 12 weeks of 450 mg green tea and 25 mg vitamin C supplementation. In contrast, Zhang et al. [24] identified only glucuronide forms of phenylvalerolactones and phenylcinnamic acids in plasma.

In a study by Zhao et al. [52], a drink supplemented with catechin and epicatechin was inoculated with gut microbiota from healthy human participants and incubated at 37 °C for 24 h under anaerobic conditions. After this treatment, all catechin and epicatechin was metabolised and significant production of gallic, 3,4-dihydroxybenzoic and homovanillic acids was detected. A single bacterial isolate was also able to convert catechin and epicatechin into several phenolic acid metabolites. This activity could potentially enhance the bioefficacy of polyphenols that are poorly absorbed in the upper gastrointestinal tract, generating phenolic metabolites with greater bioavailability and an extended half-life.

### 2.5. Anthocyanins, Proanthocyanidin and Ellagitannins

Raspberry, strawberry and pomegranate contain a wide variety of polyphenols, including anthocyanins and the poorly absorbed ellagitannins, which are transformed by the gut microbiota into urolithins. After consumption of red raspberry purée, three urolithins (urolithin A, urolithin A glucuronide and urolithin B glucuronide) were identified in urine. Microbial metabolites derived from anthocyanins and proanthocyanidins (benzoic acids, phenyl acids, phenylcinnamic acids and phenylvalerolactones) were also detected [24]. An increase in urolithins and the derivative urolithin A glucuronide was found after the consumption of fresh strawberries or the equivalent dose of strawberry purée [53]. The urolithins were found in urine for up to 92 h after consumption, indicating long persistence in the body after only a single dose of an ellagitannin-rich food.

Nuñez-Sánchez et al. [54] identified urolithin derivatives in plasma from 33 colorectal cancer patients after 15 days of 900 mg/day of pomegranate extract consumption. The same authors also identified 23 metabolites in urine, including ellagic acid, four methyl ellagic acid derivatives, gallic acid, two isomers of valoneic acid dilactone, gallagic acid dilactone and 15 urolithin derivatives (Appendix A).

In their pioneering work in this field, Zhang et al. [24] identified polyphenol microbial metabolites in human breast milk after chronic consumption of 125 g/day of red raspberry purée. The most relevant MPM belonged to the group of urolithins (Appendix A).

Other investigations with polyphenols have associated anthocyanins [55] and high-flavanol cocoa [56] with enhanced growth of the beneficial bacteria *Lactobacillus and Enterococcus spp*. After entering the colon, anthocyanins are fermented by the intestinal microbiota and the resulting metabolites (quercetin and phloretin derivatives; caffeoylquinic, caffeic and coumaroylquinic acids; procyanidins and catechins) may be responsible for the observed health effects in vivo [57].

Ellagitannins and ellagic acid undergo microbial metabolism in the gastrointestinal tract leading to urolithins [58], which have potential use as intake biomarkers for foods rich in those polyphenols, such as pomegranates, berries, nuts and oak-aged red wines [59]. Their promising application as functional foods depends on determination of the bacteria responsible for urolithin production, as well as elucidating the urolithin production pathways. García-Villalba et al. [60] studied the time course of urolithin production from ellagic acid by human faecal microbiota from two individuals with different urinary excretion patterns, excreting either urolithin A or isourolithin A. They found that bacteria from the *Clostridium coccoides* group played a crucial role in the production of both urolithins and the metabolic intermediates were urolithins M-5, M-6, M-7, C and E.

To further investigate the production of urolithins from ellagic acid and ellagitannins, Selma et al. [58] performed an in vitro study with a newly described species, *Gordonibacter urolithinfaciens*, isolated from the human faeces of a healthy participant. In pure cultures under anaerobic conditions, these intestinal bacteria sequentially produced urolithins M-5, M-6 and C, whereas urolithins A and B and isourolithin A were not detected. A more complete understanding of the metabolism of urolithins A and B and isourolithin A requires more investigation under the physiological conditions found in vivo.

### 2.6. Flavonols

Some studies have suggested that the clinical effects of some traditional herbal remedies might be caused by the metabolites of isorhamnetin glucoside an abundant polyphenol. Du et al. [61] supplemented a mixture of preculture bacteria and general anaerobic medium broth with isorhamnetin glucoside, and after 48 h of incubation five metabolites (isorhamnetin, kaempferol, quercetin, kaempferol glucoside and acetylated isorhamnetin glucoside) were tentatively identified. Kaempferol is widely reported to possess a range of pharmacological properties including anticancer, cardioprotective, neuroprotective, antioxidant, anti-inflammatory, antimicrobial and antiallergic activity [62]. The health benefits of quercetin include protection against several diseases such as osteoporosis, pulmonary and cardiovascular diseases and cancer [63]; antiproliferative, antiatherosclerotic and neuroprotective effects are also described [64].

## 3. Other Food Intervention Studies with Mixed Polyphenols

Grapes and grape-derived products such as wine are rich in phenolic compounds, particularly flavonoids, anthocyanins, proanthocyanidins, procyanidins, phenolic acids and stilbenes. The various beneficial health effects associated with their consumption seem due to this wide variety of bioactive compounds [65].

Boto-Ordóñez et al. [20] designed an open, randomised, crossover, controlled intervention trial where nine healthy men followed three 20-day interventions with dealcoholised red wine, red wine or gin. The aim was to evaluate the association between changes in the concentration of intestinal bacteria and urinary microbial phenolic acids. *Bifidobacterium* was significantly correlated with differences in 4-hydroxybenzoic, syringic, *p*-coumaric and homovanillic acids, which are metabolic products of anthocyanin degradation. Although the correlation was statistically significant, the authors highlight the difficulty of establishing whether these metabolites are derived only from anthocyanin or also from other sources. Homovanillic acid, for instance, could also be derived from ferulic acid, *p*-coumaric acid from dehydroxylation of caffeic acid, and syringic acid from gallic acid.

Functional beverages based on grape extracts were used to observe modifications in the urinary metabolome. A study by Khymenets et al. [65,66] involved 31 healthy participants who consumed a drink containing grape skin extract or a placebo control for 15 days. Urine was collected on the first day during the first four postprandial hours, and for 24 h on the last day of the intervention. Among 18 metabolites identified after consumption of grape skin extract, 4-hydroxyhippuric acid was excreted in a high concentration, due to both *p*-hydroxybenzoic acid glycination and microbial metabolic activities. Hydroxydimethoxybenzoic acid glucuronide, a conjugate of syringic and dimethylgallic acids, was also identified. Other metabolites found were syringic acid, 3-methylgallic acid, vanillic acid glucuronide and vanilloylglycine, which could derive directly from the grape extract or have been produced by the microbiota from hydroxycinnamates and flavan-3-ols. Dihydrosinapic acid glucuronide was identified as a metabolite of sinapic acid, the most abundant hydroxycinnamate found in red grapes and derived products. Finally, the authors found eight metabolites of first-stage flavan-3-ol microbial metabolism, namely derivatives of hydroxy(dihydroxyphenyl)valeric acid (two glucuronides and sulphate), dihydroxyphenyl (two glucuronides), hydroxymethoxyphenyl (two glucuronides) and hydroxyphenylvalerolactones (one glucuronide) [65] (Appendix A).

In a study by Sasot et al. [66], a beverage made with grape pomace was administered to 12 healthy participants, with urine collected before intake and 24 h afterwards. Seventy phenolic metabolites were identified through LC-ESI-LTQ-Orbitrap-MS analysis (see Appendix A). Some of these metabolites hydroxybenzoic acids, syringic acid, hydroxyhippuric acid and hydroxyphenylpropionic acids were also found by Khymenets et al. [65], who also identified two phenolic compounds derived from intestinal microflora, 4-hydroxy-5-(3,4-dihydroxyphenyl)valeric acid and 5-(3´,4´-dihydroxyphenyl)-γ-valerolactone. The former had previously been described as a gut microbiota-derived metabolite produced after grape intake, while the latter is the most abundant metabolite of (epi)catechin and procyanidin B degradation and can be metabolised into simple phenolic acids such as 3-hydroxyphenylpropionic acid, 3-hydroxyhippuric acid and 3-hydroxybenzoic acid.

Most clinical trials in this area have focused on the impact of red wine on gut microbiota, likely because of available evidence for its preventive effects against chronic diseases [50,67,68]. Red wine polyphenols may modulate the profile and activity of gut microbiota, increasing the growth of probiotics bacteria such as *Bifidobacterium* and decreasing non-beneficial bacteria. This could be due to the antimicrobial characteristics of red wine, which affect host microbiota interaction [69]. In line with this, studies have reported changes in the profile of MPM in human faeces after moderate intake of red wine polyphenols [50,67,68]. In a targeted analysis by liquid chromatography coupled to mass spectometry (LC-MS), an increase in MPM including 3,5-dihydroxybenzoic acid, 3-methylgallic acid, *p*-coumaric acid, phenyl propionic acid, protocatechuic acid, vanillic acid, syringic acid and 4-hydroxy-5-(phenyl)valeric acid was observed after moderate wine intake [50,68] (Appendix A). Final products of MPM may derive from the metabolism of oligomers and polymers of flavan-3-ols and procyanidins in the colon [68]. Furthermore, a non-targeted analysis after wine intake revealed the intra and inter-individual variability of faecal polyphenol metabolites [68].

Tomato and tomato-based products are rich in phenolic acids (homovanillic acid hexoside, 5-caffeoylquinic acid, caffeic acid hexoside-I and ferulic acid), flavanones (naringenin) and flavonols (rutin). In an open, controlled, randomised and crossover feeding trial with 40 healthy participants, MPM were analysed in urine after consumption of tomato and tomato sauces. A wide variety of metabolites and their glucuronide and sulphate conjugates were detected, namely 3-(4-hydroxyphenylpropionic) acid, 3-(3-hydroxyphenyl)propionic acid, 4-hydroxyphenylpropionic acid, 3-phenylpropionic acid, 4-hydroxyhippuric acid, 3,4-dihydroxyphenylacetic acid, 3-hydroxyphenylacetic acid, and phenylacetic acid (see Appendix A). Higher concentrations were obtained for the glucuronide and sulphate metabolites than for their aglycones [70].

After supplying an almond skin extract to 24 healthy participants, Llorach et al. [71] identified 34 metabolites in urine, classified as conjugates of flavonoids, hydroxyphenylvalerolactone, 4-hydroxy-5-(phenyl)-valeric acid, hydroxyphenylpropionic acid, hydroxyphenylacetic acid and other phenolic acids.

Table 1 summarises the main families of MPM found in biological fluids. An extended version of the table is available in the Appendix A.

## 4. Concluding Remarks

A growing number of health benefits are being attributed to the gut microbiota, although research on polyphenol-microbe interactions is hampered by the immense variability in microbiota not only between individuals but also within the same person at different life-stages and states of wellbeing. Cardiovascular health and protection against oxidative and inflammatory states have been associated with the consumption of some polyphenols, which are, in turn, metabolised by the microbiota into more active molecules. In some cases, the interaction between polyphenols and the microbiota is so deep that a particular polyphenol can induce the growth of a beneficial bacterial population. Attempts to decipher the role of this microscopic population in the effects on the human body of bioactive compounds such as polyphenols have used distinct approaches.

In vitro studies are useful in early stages of research because they provide direct evidence of polyphenol modification and metabolite production without interference of the human physiology. They also allow the study of the probiotic effect that polyphenols might have on bacterial cultures. Models of the human gut were developed years ago and have provided a close reproduction of what actually happens within the intestine. However, these models do not replicate the whole gastrointestinal apparatus, and thus, the impact of initial stages of digestion, such as mastication, cannot be assessed. Another limitation is the difficulty in studying reactions that take place once absorption has occurred.

Human clinical studies provide more exhaustive results, but their interpretation is complex. Only part of the polyphenol content of foods reaches the large intestine, where they may undergo a number of reactions, depending on the location. Furthermore, the microbiota composition and metabolism vary from person to person and are not static. The wide range of processes susceptible to genetic variability, which include modification by the microbiota, absorption, phase II conjugation, circulation, activity and excretion, represent a challenge for researchers in this field. Furthermore, the potentially health-modifying molecules that reach the bloodstream are present in such low concentrations and are structurally so varied that it is difficult to reach firm conclusions. This is not to mention the intricacy of biological matrices, which have to be cleaned up and the compounds extracted to obtain neater samples to analyse.

The use of HPLC coupled with MS detectors has facilitated the detection, identification and quantification of polyphenols and their metabolites. However, the variety of methodologies used by research groups, together with the immense number of metabolites, makes gathering and organising all the metabolites and metabolic pathways a difficult task. One limitation is the targeting of the studies: research groups are usually focused on one polyphenol group or type. They target their studies, leaving out of their investigations metabolites that come from polyphenols present in foods, although maybe in small quantities. This is why studies using the same base food find different metabolites in the samples. This issue complicates the comparison between studies and the construction of a global vision around microbiota action.

To sum up, the possible beneficial effects of polyphenols seem partly dependent on their microbiota-derived metabolites, as these are the molecules reported to normally cross the gut lining. Despite the complexity of research in this field, advances in sample extraction and molecule detection have made data analysis more manageable. Further studies are required to create a comprehensive map of polyphenol metabolism and the effect of the generated molecules on human health.

## Figures and Tables

**Figure 1 nutrients-11-02725-f001:**
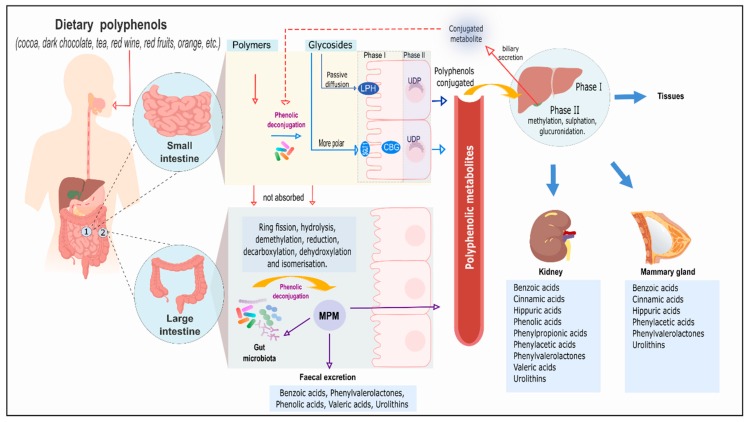
Metabolic pathways of polyphenols produced by the human gut microbiota. LPH: lactase-phlorizin hydrolase; CBG: cytosolic β-glucosidase; UDP: uridine 5’-diphospho-glucuronosyltransferase; MPM: microbial phenolic metabolites.

**Table 1 nutrients-11-02725-t001:** Principal MPM groups found in biological fluids after feeding trials in humans.

GUT MICROBIOTA METABOLITES	FOOD SOURCE	SAMPLE TYPE	REFERENCES
Benzoates	Red raspberry	Breast milk, plasma, urine	[24]
Hydroxybenzaldehydes	Red raspberry	Breast milk, plasma, urine	[24]
Hydroxybenzoic acids	Almond skin, capsule of epicatechin and procyanidin B1, cocoa, coffee, grape extract, orange juice, pomegranate extract, red raspberry, red wine, tea, tomato sauce	Blister fluids, breast milk, colon tissues, faeces, plasma, skin biopsies, urine	[20,24,26,27,28,35,42,43,45,50,51,54,65,66,68,70,71]
Hydroxycinnamic acids	Capsule of epicatechin and procyanidin B1, cocoa, coffee, grape extract, orange juice, red raspberry, red wine, tomato sauce	Breast milk, faeces, plasma, urine	[20,24,26,28,35,42,43,45,50,66,70,71]
Hydroxycoumarins	Pomegranate extract, red raspberry, strawberry	Breast milk, colon tissues, plasma, urine	[24,53,54]
Hydroxyphenylacetic acids	Almond skin, capsule of epicatechin and procyanidin B1, cocoa, coffee, grape extract, orange juice, red raspberry, red wine, tomato sauce	Breast milk, faeces, plasma, urine	[20,24,26,27,28,35,42,43,45,48,50,66,68,70,71]
Hydroxyphenylpentanoic acids	Almond skin, capsule of epicatechin, dark chocolate, grape extract, red wine, tea	Faeces, plasma, urine	[43,44,48,50,65,66,67,68,71]
Hydroxyphenylpropanoic acids	Almond skin, capsule of epicatechin and procyanidin B1, cocoa, coffee, dark chocolate, grape extract, orange juice, red raspberry, red wine, tea, tomato sauce	Blister fluids, breast milk, faeces, plasma, urine	[20,24,26,27,28,35,42,43,44,45,48,50,51,65,66,67,68,70,71]
Valerolactones	Almond skin, capsule of epicatechin and procyanidin B1, cocoa, dark chocolate, grape extract, red raspberry, red wine, tea	Breast milk, faeces, ileostomy fluids, plasma, urine	[24,25,43,44,45,46,48,49,51,65,66,67,68,71]
Others	Coffee, grape extract, grapevine-shoot supplement, orange juice, red raspberry, tea	Ileostomy fluids, plasma, urine	[19,24,26,28,35,48,49,66]

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
