# Peer review of "Microbial Phenolic Metabolites: Which Molecules Actually Have an Effect on Human Health?"

_nutrients, 2019, doi:10.3390/nu11112725_

Round 1

Reviewer 1 Report

Although readers cannot imagine each structure only by each name, this second version of the paper is a great improvement. Because authors summarized it according to the biologically active compounds.

Author Response

The authors want to thank the reviewer for his/her time to review our manuscript.

Reviewer 2 Report

This review summarized metabolites of dietary polyphenols, which are classified with category, produced by host metabolisms and gut microbiota. Comprehensive information on metabolites of dietary polyphenols derived from intestinal microorganisms is useful to readers. Refer general and individual comments.

General comments

1, Some sentences are hardly understandable in introduction and conclusion remarks, which may be language problems, for example L.69-71, L.760-764. Please check again.

2, Phenolic compounds and gut-microbe interaction is main subject of this review and very important aspects for beneficial effects of dietary polyphenols. However, the descriptions of effects of dietary polyphenols on changes in gut microbiota populations are scattered throughout the manuscript. It is better to write these together and discuss on them.

Individual comments

1, L.54: What dose mean “imbalance nutrient uptakes”. Authors should reward and rephrase this part. The word “imbalance” is a specific term for amino acid nutrition, and mean of this phrase is unclear even though “imbalance” is replaced by “unbalance”.

2, L.232: “Dihydroxyvalerolactone sulphate” may be “Dihydroxyphenylvalerolactone sulphate”, though “Dihydroxyvalerolactone sulphate” is found in abstract of reference 46.

3, L411: “Boto et al [20]” should be “Boto-Ordonez et al [20]”.

4, L508-513: What are parent polyphenols in tomato for described metabolites?

5, L:766-777: This sentence is final conclusion of this review manuscript, however a word “mostly” in this sentence is better to reword “partly” because it is less evidence.

Author Response

Responses to reviewer 2

The authors want to thank the reviewer for his/her time to review our manuscript.

Comment 1.  Some sentences are hardly understandable in introduction and conclusion remarks, which may be language problems, for example L.69-71, L.760-764. Please check again.

Response: Thanks for your comment. We have checked and reviewed the English throughout the manuscript.

Comment 2.  Phenolic compounds and gut-microbe interaction is main subject of this review and very important aspects for beneficial effects of dietary polyphenols. However, the descriptions of effects of dietary polyphenols on changes in gut microbiota populations are scattered throughout the manuscript. It is better to write these together and discuss on them.

Response: We thank the reviewer for this comment. We have written all these descriptions together (lines 380-384).

Comment 3.  L.54: What does mean “imbalance nutrient uptakes”. Authors should reward and rephrase this part. The word “imbalance” is a specific term for amino acid nutrition and mean of this phrase is unclear even though “imbalance” is replaced by “unbalance”.

Response: Thanks for your suggestion. We have replaced imbalance by unbalance (line 54).

Comment 4.   L.232: “Dihydroxyvalerolactone sulphate” may be “Dihydroxyphenylvalerolactone sulphate”, though “Dihydroxyvalerolactone sulphate” is found in abstract of reference 46.

Response: Thanks for the comment. The correct compound should be dihydroxyphenylvalerolactone sulphate (line 215).

Comment 5.   L411: “Boto et al [20]” should be “Boto-Ordonez et al [20]”.

Response: Thanks for your comment. We have changed it (line 311).

Comment 6.  L508-513: What are parent polyphenols in tomato for described metabolites?

Response: We thank the reviewer for this comment. We have indicated this in lines 359-361.

Comment 7.   L:766-777: This sentence is final conclusion of this review manuscript, however a word “mostly” in this sentence is better to reword “partly” because it is less evidence.

Response: Thanks for the comment. We have changed the word (line 409).

This manuscript is a resubmission of an earlier submission. The following is a list of the peer review reports and author responses from that submission.

Round 1

Reviewer 1 Report

There are many problems in the review manuscript of “Microbial phenolic metabolites: which molecules actually have effect on human health?” as given below.

(1) This review is nothing but a meaningless list of the related references.

(2) Authors should cite the chemical structure of crucial compounds in the manuscript to help understanding for readers

(3) As this manuscript is involved in ADME of biologically active compounds (phenolic metabolites), authors should describe the metabolic route using each chemical structure.

(4)The same compound in the manuscript should be abbreviated by figures, not by the full name.

Reviewer 2 Report

See attachment.
